# Atrial TRPM2 Channel-Mediated Ca^2+^ Influx Regulates ANP Secretion and Protects Against Isoproterenol-Induced Cardiac Hypertrophy and Fibrosis

**DOI:** 10.3390/cells15010024

**Published:** 2025-12-22

**Authors:** Tomohiro Numata, Hideaki Tagashira, Kaori Sato-Numata, Meredith C Hermosura, Fumiha Abe, Ayako Sakai, Shinichiro Yamamoto, Hiroyuki Watanabe

**Affiliations:** 1Department of Integrative Physiology, Graduate School of Medicine, Akita University, Akita 010-8543, Japan; htagashira@med.akita-u.ac.jp (H.T.); satokao@med.akita-u.ac.jp (K.S.-N.); f_abe@hos.akita-u.ac.jp (F.A.); a.sakai@med.akita-u.ac.jp (A.S.); 2John A. Burns School of Medicine, Honolulu, HI 96813, USA; hermosur@hawaii.edu; 3Faculty of Pharmaceutical Sciences, Teikyo Heisei University, Tokyo 164-8530, Japan; s.yamamoto@thu.ac.jp; 4Department of Cardiovascular Medicine, Akita University Graduate School of Medicine, Akita 010-8543, Japan; hirow@doc.med.akita-u.ac.jp

**Keywords:** TRPM2, atrial natriuretic peptide (ANP), calcium signaling, cardiac hypertrophy, fibrosis, beta-adrenergic stress, cardiac hormones

## Abstract

**Highlights:**

**What are the main findings?**
TRPM2 is functionally enriched in atrial cardiomyocytes and drives stress-evoked Ca^2+^ influx that triggers ANP secretion.TRPM2 deficiency exacerbates ISO-induced hypertrophy, fibrosis, and systolic dysfunction, with blunted *Nppa*/ANP induction; exogenous ANP rescues these phenotypes.

**What are the implications of the main findings?**
TRPM2 functions as an upstream Ca^2+^-dependent regulator of the cardiac natriuretic peptide system, extending its known roles beyond oxidative-stress sensing.Therapeutic augmentation of the ANP axis (e.g., ANP administration, neprilysin inhibition) or targeting TRPM2 may mitigate pathological cardiac remodeling in hypertension and heart failure.

**Abstract:**

Transient receptor potential melastatin 2 (TRPM2) channel is a Ca^2+^-permeable, redox-activated cardiac ion channel protective in ischemia–reperfusion, but whether it regulates atrial endocrine output under stress is unclear. Here, we investigated whether TRPM2 contributes to the atrial natriuretic peptide (ANP) response during β-adrenergic stimulation. We compared how male C57BL/6J wild-type (WT) and TRPM2 knockout (TRPM2^−/−^) mice (8–12 weeks old) respond to β-adrenergic stress induced by isoproterenol (ISO) using echocardiography, histology, RT-PCR, electrophysiology, Ca^2+^ imaging, ELISA, and atrial RNA-seq. We detected abundant Trpm2 transcripts in WT atria and measured ADP-ribose (ADPr)-evoked currents and hydrogen peroxide (H_2_O_2_)-induced Ca^2+^ influx characteristic of TRPM2; these were absent in TRPM2^−/−^ cells. Under the ISO-induced hypertrophic model, TRPM2^−/−^ mice developed greater cardiac hypertrophy, fibrosis, and systolic dysfunction compared with WT mice. Atrial bulk RNA-seq showed significant induction of *Nppa* (ANP precursor gene) in WT + ISO, accompanied by higher circulating ANP; TRPM2^−/−^ + ISO showed blunted *Nppa* and ANP responses. ISO-treated TRPM2^−/−^ mice exhibited more blunt responses, in both *Nppa* transcripts and circulating ANP levels. Exogenous ANP attenuated ISO-induced dysfunction, hypertrophy, and fibrosis in TRPM2^−/−^ mice, suggesting that TRPM2 is needed for the cardioprotective endocrine response via ANP to control stress-induced β-adrenergic remodeling.

## 1. Introduction

Cardiac hypertrophy and fibrosis are hallmark features of pathological remodeling in response to chronic β-adrenergic stimulation, as seen in conditions such as hypertension and heart failure [1,2,3,4,5]. In these settings, reactive oxygen species (ROS) act as active amplifiers of maladaptive signaling, arising from mitochondrial and enzymatic sources, and promote pro-hypertrophic and pro-fibrotic pathways [6,7,8,9,10]. Atrial natriuretic peptide (ANP), secreted primarily by atrial cardiomyocytes, plays a pivotal endocrine role in attenuating cardiac remodeling by promoting natriuresis, vasodilation, and antifibrotic signaling [11,12,13]. Accordingly, regulated ANP secretion is a critical adaptive mechanism; however, the upstream molecular triggers that govern ANP release under stress conditions remain incompletely understood. ISO-induced β-adrenergic stimulation is a widely used model that recapitulates key features of human catecholamine-driven cardiac remodeling, including ROS elevation, neurohumoral activation, hypertrophy, and fibrosis, making it suitable for mechanistic studies of stress-induced endocrine responses.

The transient receptor potential melastatin 2 (TRPM2) channel is a Ca^2+^-permeable, non-selective cation channel activated by oxidative stress, ADPr, and hypertonicity [14,15,16,17]. TRPM2 is expressed in a wide variety of cells and tissues, including the cardiovascular system, where it has been implicated in redox/inflammatory signaling, cardioprotection against ischemia–reperfusion (I/R) injury [18], and in vascular/endothelial biology [19,20]. Mechanistically, oxidative DNA damage can activate PARP/PARG, elevating free ADPr that binds to the TRPM2 NUDT9-H domain to gate the channel and facilitate Ca^2+^ influx [15,21,22]. Studies using ventricular myocytes have provided evidence ascribing the protective effects of TRPM2 to Ca^2+^ entering the channel [23] and causing the phosphorylation of Pyk2, which in turn translocates to the mitochondria, thereby improving mitochondrial bioenergetics and maintaining cardiac health [24,25]. The C57BL/6J background is widely used in cardiovascular research and enables reproducible assessment of hypertrophic and fibrotic responses, while TRPM2^−/−^ mice provide a genetic approach to dissect redox-sensitive Ca^2+^ influx pathways relevant to ANP regulation.

In preliminary studies, we found robust expression of TRPM2 transcripts in atrial cardiomyocytes. Given that the cardiac hormone ANP is primarily produced by atrial cardiomyocytes, we wondered if Ca^2+^ entering through TRPM2 channels influences the production and/or secretion of ANP, thereby providing an upstream Ca^2+^ signal that triggers ANP release in response to β-adrenergic stress. We hypothesize that TRPM2-mediated Ca^2+^ entry in atrial cardiomyocytes contributes to stress-evoked ANP release. The specific objective of this study was to determine whether TRPM2-mediated Ca^2+^ entry is required for the appropriate induction and secretion of ANP during β-adrenergic stress, and to assess how TRPM2 deficiency affects the resulting cardiac remodeling. We tested this hypothesis by investigating the role of TRPM2 in atrial endocrine signaling and its relationship to cardiac remodeling using WT and TRPM2 knockout (TRPM2^−/−^) mice subjected to isoproterenol (ISO)–induced β-adrenergic stress. This framework guided our experimental design across molecular, cellular, and in vivo analyses. Guided by this rationale, we performed integrated molecular, cellular, and in vivo experiments to determine how TRPM2 contributes to atrial endocrine signaling and stress-induced cardiac remodeling.

## 2. Materials and Methods

Study design: For in vivo experiments, the following treatment groups were compared: (1) Control (vehicle-treated), (2) ISO-treated (30 mg/kg/day), (3) ANP-treated alone, and (4) ISO + ANP co-treated mice. For genotype comparison, the same groups were examined in WT and TRPM2^−/−^ mice. A single mouse served as the experimental unit for all in vivo analyses. For in vitro assays (isolated atrial myocytes, ventricular myocytes, and NMVMs), each culture well or each individual isolated cell (for electrophysiology and Ca^2+^ imaging) was treated as one experimental unit.

### 2.1. Animals and In Vivo Treatment Protocol

All experimental procedures were conducted in the Akita University SPF animal facility. Mice were allowed to acclimatize for at least 7 days prior to the start of experiments. ISO (30 mg/kg/day, i.p.) was administered once daily for 7 or 21 days to induce pathological cardiac stress, a well-established model of cardiac hypertrophy and fibrosis. ANP (400 µg/kg, s.c., twice daily) was administered during ISO treatment to determine whether enhancing natriuretic peptide signaling mitigates ISO-induced cardiac dysfunction. These procedures and dosing regimens were selected based on previous evidence demonstrating reproducible induction of cardiac hypertrophy and robust activation of endogenous ANP signaling.

Male WT and TRPM2^−/−^ mice on a C57BL/6J background (8–12 weeks old; body weight approximately 22–28 g) were used. TRPM2^−/−^ mice were originally generated and provided by Dr. Yasuo Mori (Kyoto University) and had been backcrossed onto the C57BL/6J background. WT littermates from the same breeding colonies served as controls. All mice were bred and maintained under specific pathogen-free conditions in the animal facility of Akita University, under controlled environmental conditions (25 ± 1 °C; 12 h light/dark cycle) with ad libitum access to standard chow and water. Mice were housed in individually ventilated cages (2–5 mice per cage) with autoclaved wood-chip bedding that permitted natural burrowing and nest-building. Animals were maintained under SPF conditions with routine daily husbandry performed by trained staff. Genotypes (TRPM2 WT or TRPM2^−/−^) were confirmed by PCR analysis of tail DNA. All animals were experimentally naïve prior to enrollment in the present study. To induce pathological cardiac stress, mice received once-daily intraperitoneal injections of isoproterenol (ISO; 30 mg/kg/day) for 7 or 21 consecutive days [26]; age-matched controls were administered volume-matched vehicle. For ANP supplementation, a subset of mice received atrial natriuretic peptide (ANP; 400 µg/kg, s.c.) twice daily for 7 days, administered subcutaneously on the same daily schedule during the ISO protocol; corresponding controls received vehicle. All experimental procedures complied with institutional guidelines and were approved by the Animal Ethics Committee of Akita University (approval nos. a-1-0412 and b-1-0408). Mice were randomly assigned to treatment groups, and outcome assessments were performed by investigators blinded to genotype and treatment. Sample size; for in vivo experiments, a total of 38 WT mice and 48 TRPM2^−/−^ mice were allocated across the four treatment groups (Control, ISO, ANP, ISO + ANP), depending on the specific endpoint (echocardiography, histology, ELISA, or RNA analyses), as detailed in the figure legends. In total, 86 mice were used in this study across all treatment groups and analyses. Sample size was determined based on our previous studies using ISO-induced cardiac hypertrophy models and TRPM2^−/−^ mice, which demonstrated that group sizes of 6–8 animals provide adequate statistical power to detect physiologically relevant differences in hypertrophy, fibrosis, and ANP responses. Inclusion and exclusion criteria; Criteria were established a priori. For in vivo experiments, mice were included if they completed the full treatment protocol and showed no signs of severe distress. According to institutional animal care guidelines, mice were excluded if they exhibited any of the following: (1) body weight loss exceeding 10% from baseline, (2) labored breathing or impaired mobility, or (3) visible wounds, ulceration, or signs of infection. No mice met these exclusion criteria during the study, and therefore no animals were removed from any treatment group. For in vitro experiments (electrophysiology, Ca^2+^ imaging, and NMVM hypertrophy assays), only quiescent, rod-shaped cardiomyocytes with clear striations and stable baseline signals were included. Cells exhibiting contracture, membrane blebbing, loss of striations, unstable access resistance (>20% change during recording), or baseline Ca^2+^ ratio drift were excluded according to predefined criteria. No animals, cell samples, or data points were excluded for technical or biological reasons, unless noted in the figure legends. The exact n for each experimental group, including animals and independent cell preparations, is reported in the corresponding figure legends. Randomization; Mice were randomly assigned to treatment groups (Control, ISO, ANP, ISO + ANP) using simple random allocation. Assignment was carried out by an investigator not involved in data collection or outcome assessment. To minimize potential confounders, mice were housed in mixed-genotype and mixed-treatment-condition cages, when possible, with cage location on the rack rotated weekly to avoid positional bias. The order of echocardiography, tissue collection, histology, and biochemical measurements was also randomized across groups to prevent sequence effects. For in vitro experiments (electrophysiology, Ca^2+^ imaging, and NMVM assays), cells from each mouse were processed in alternating order between genotypes and treatment conditions on each experimental day to reduce batch and timing confounders. If cage effects or measurement-order effects could not be fully controlled for technical reasons (e.g., limited imaging slots or isolation yield), these factors were balanced across groups rather than fixed to a single condition. Blinding; Group allocation was performed by an investigator who was not involved in data collection or outcome assessment. During the conduct of the in vivo experiments (administration of ISO, ANP, echocardiography, and tissue collection), investigators were aware of the treatment protocol but remained blinded to genotype whenever possible, as genotyping information was coded prior to experiments. Outcome assessments, including echocardiographic analysis, histology and fibrosis quantification, ELISA measurements, and all in vitro analyses (electrophysiology, Ca^2+^ imaging, and NMVM hypertrophy assays), were performed by investigators blinded to both genotype and treatment group. Image analysis and morphometric quantification were carried out using coded file names to maintain blinding. Data processing and statistical analysis were conducted using de-identified datasets, and group codes were revealed only after all analyses were completed. Outcome measures: the primary outcome measure for this study was the degree of cardiac hypertrophy and dysfunction induced by ISO, assessed by (1) heart weight-to-body weight (HW/BW) ratio and (2) echocardiographic parameters (fractional shortening and ejection fraction). These measures were selected a priori and used in determining the required sample size based on previous studies using the ISO-induced hypertrophy model. Secondary outcome measures included: Histological assessment of myocardial fibrosis (Masson’s trichrome staining and quantitative fibrosis fraction). ANP-related outcomes, including plasma ANP concentration (ELISA), atrial *Nppa* mRNA expression (qPCR), and Npr1 expression. TRPM2 channel activation assessed by patch clamp in isolated atrial myocytes (ADPr/H_2_O_2_-evoked whole-cell currents). Intracellular Ca^2+^ dynamics measured by fura-2 Ca^2+^ imaging in atrial myocytes following oxidative stimulation. Hypertrophic responses in neonatal mouse ventricular myocytes (NMVMs), quantified by cross-sectional area. All outcome measures were defined before data collection began, and no exploratory endpoints were introduced after the start of the study.

Animal care and monitoring: throughout all in vivo procedures, animals were monitored at least once daily by trained personnel to assess general condition, mobility, grooming behavior, and food/water intake. ISO injections were performed using refined handling and injection techniques to minimize stress and discomfort. To reduce procedure-related distress, all echocardiographic measurements were conducted under light ketamine/xylazine anesthesia as approved in the institutional protocol. No analgesics were required, as none of the procedures were expected to cause sustained pain. No unexpected adverse events occurred during the study. A small, transient reduction in activity was occasionally observed immediately after ISO injection, but all mice recovered within minutes and displayed normal behavior thereafter. Humane endpoints were predefined according to institutional guidelines. Animals were removed from the study if any of the following were observed: (1) >10% loss of body weight from baseline, (2) labored breathing, impaired mobility, or persistent lethargy, or (3) visible wounds, ulceration, or signs of infection. None of the mice reached these humane endpoints, and no animals required early euthanasia.

### 2.2. Echocardiography

Cardiac function was evaluated by transthoracic echocardiography using the Vevo 770 system (FUJIFILM VisualSonics Inc., Toronto, ON, Canada) under ketamine (50 mg/kg) and xylazine (5 mg/kg) anesthesia. Fractional shortening (FS%), ejection fraction (EF%), left ventricular end-systolic diameter (LVESD), and end-diastolic diameter (LVEDD) were calculated from M-mode tracings.

### 2.3. Histology and Morphometry

At the endpoint, mice were euthanized by cervical dislocation, and hearts were excised for histological analysis. Heart weight-to-body weight (HW/BW) ratios were determined. Paraffin-embedded sections were stained with hematoxylin–eosin (HE) and Masson’s trichrome (MT) to evaluate myocardial hypertrophy and fibrosis. Images were acquired using a BZ-X800 inverted microscope (KEYENCE, Tokyo, Japan). Fibrosis quantification. Masson’s trichrome (MT) images were analyzed in Adobe Photoshop (Adobe Systems, San Jose, CA, USA) using a pre-specified, uniform workflow applied to all samples. First, the background signal was minimized by applying the Image Subtract function using a blank-area region of interest sampled on each slide. Next, collagen-positive regions were segmented using the Color Range selection of the MT blue component, with a fixed tolerance determined a priori and held constant across all images. Total tissue area (myocardium only) was obtained by excluding background and non-tissue spaces (e.g., cavities) using thresholding with minor manual cleanup under the same settings. The fibrosis fraction was calculated as the number of blue-positive pixels divided by the total number of tissue pixels, expressed as a percentage.

### 2.4. Isolation of Atrial and Ventricular Cardiomyocytes

Single cardiomyocytes were isolated by retrograde aortic perfusion (Langendorff method) using Ca^2+^-free Tyrode’s solution, followed by enzymatic digestion with collagenase type II (Worthington, Lakewood, NJ, USA). After digestion, the atria and ventricles were dissected and gently triturated in enzyme-containing solution, and the cell suspensions were filtered and allowed to settle by gravity. Cells were then transferred to the recording buffer. Only quiescent, rod-shaped myocytes with apparent striations were selected. Electrophysiological recordings and Ca^2+^ imaging were performed within 8 h of isolation.

### 2.5. Whole-Cell Patch-Clamp Electrophysiology

Whole-cell recordings were obtained from isolated atrial myocytes of WT and TRPM2^−/−^ mice at room temperature (22–27 °C). Patch electrodes were pulled from borosilicate glass capillaries using a P-97 puller (Sutter Instrument, Novato, CA, USA) and had tip resistances of 2–5 MΩ (typically 3–5 MΩ). Currents were recorded in the whole-cell configuration using either an EPC-9 amplifier (HEKA Elektronik, Lambrecht, Germany) or an Axopatch 200B amplifier (Axon Instruments/Molecular Devices, Union City, CA, USA). Signals were low-pass filtered at 5 kHz (four-pole Bessel), digitized at 10 or 20 kHz, and acquired/controlled with PULSE v8.8 (HEKA) or pCLAMP v10.0.2 (Molecular Devices). Series resistance was compensated for 70–80% to minimize voltage errors. Only quiescent, rod-shaped cells with apparent striations were analyzed, and current amplitudes were normalized to membrane capacitance (pA/pF).

External (bath) solution (pH 7.2 with NaOH; 320 mOsm): NaCl 145 mM, MgCl_2_·6H_2_O 1.2 mM, CaCl_2_ 0.2 mM, HEPES 11.5 mM, D-glucose 10 mM. Internal (pipette) solution (pH 7.2 with CsOH; 300 mOsm adjusted with D-mannitol): Cs-hydroxide 105 mM, aspartate 105 mM, CsCl 40 mM, MgCl_2_ 2 mM, CaCl_2_ 2.789 mM, K_4_-BAPTA 5 mM, Na_2_-ATP 2 mM, HEPES 5 mM; ADPr 0.5 mM was included where indicated.

Cells were held at 0 mV and subjected to a ramp protocol consisting of brief square steps to +100 mV (10 ms) and −100 mV (10 ms), followed by a linear ramp from +100 to −100 mV over 100 ms, repeated every 10 s. TRPM2 activity was assessed by comparing I–V relationships recorded with versus without 0.5 mM ADPr in the pipette in WT and TRPM2^−/−^ atrial myocytes.

### 2.6. Calcium Imaging

Intracellular Ca^2+^ was monitored in freshly isolated atrial myocytes using the ratiometric indicator fura-2 AM (Dojindo Laboratories, Kumamoto, Japan). Cells were incubated with fura-2 AM (5 µM) for 30 min at 37 °C, rinsed for 15 min at room temperature.

Recordings were obtained on an inverted epifluorescence microscope (IX81, Olympus Corp., Tokyo, Japan) equipped with a xenon arc lamp and monochromator (Polychrome IV, TILL Photonics GmbH, Gräfelfing, Germany) and a cooled CCD camera (ORCA series, Hamamatsu Photonics, Shizuoka, Japan). Fura-2 was excited alternately at 340 and 380 nm, and emission was collected through a 510 nm long-pass filter. Image acquisition and ratio calculations (F340/F380) were performed with MetaFluor software (version 7.6; Molecular Devices, San Jose, CA, USA) with fixed exposure across conditions.

All measurements were carried out in Tyrode’s solution containing (in mM) NaCl 145, MgCl_2_·6H_2_O 1.2, CaCl_2_ 0.2, HEPES 11.5, D-glucose 10; pH 7.2 with NaOH, 320 mOsm. After establishing a stable baseline (≥60 s), cells were exposed to H_2_O_2_ (oxidative stimulus; concentration indicated in figure legends) according to the experimental protocol. Changes in Ca^2+^ were quantified as ΔRatio = peak F340/F380 − baseline F340/F380. Only quiescent, rod-shaped cells with stable baselines were included.

### 2.7. Primary Culture of Neonatal Mouse Ventricular Myocytes (NMVMs) and Hypertrophy Assay

Neonatal ventricular myocytes were isolated from 1–3-day-old C57BL/6J WT and TRPM2^−/−^ pups using the Pierce Cardiomyocyte Isolation Kit (Thermo Fisher Scientific, Waltham, MA, USA). Cells were cultured in DMEM supplemented with 10% fetal bovine serum, followed by serum-free conditions before stimulation. After 48 h, hypertrophy was induced with isoproterenol (ISO, 10 µM, 24–72h) in the absence or presence of recombinant mouse ANP (0.1 µM; Peptide Institute, Osaka, Japan).

For morphometry, cells were fixed in 4% paraformaldehyde (FUJIFILM Wako Pure Chemical, Osaka, Japan) for 5 min, and the fixed cells were incubated with rhodamine-phalloidin reagent (Abcam, Cambridge, UK) in PBS containing 1% BSA (FUJIFILM Wako) for 1 h at room temperature. Fluorescence images were acquired using a BZ-X800 microscope (KEYENCE, Tokyo, Japan) with identical exposure and acquisition parameters across conditions.

Cell cross-sectional area (CSA) was quantified from phalloidin images using ImageJ (version 1.53k; NIH). For each condition, >100 cells were analyzed from ≥3 independent isolations (biological replicates); the analyst was blinded to genotype and treatment.

### 2.8. RNA-Seq

Total RNA was extracted from mouse atria and ventricles. For each genotype/treatment condition (WT, TRPM2^−/−^, WT + ISO, TRPM2^−/−^ + ISO), atria from three mice were pooled at equal tissue mass to generate one composite RNA sample, yielding four composite libraries (one per condition). Library construction and sequencing were performed by Macrogen Japan Corp. (Kyoto, Japan). Stranded poly(A)+ libraries were prepared and sequenced on an Illumina NovaSeq X (Illumina, San Diego, CA, USA), generating paired-end 101 bp FASTQ reads. Pre-processing removed rRNA-derived reads using SortMeRNA v4.3.7 with rRNA references extracted from GCF_000001405.26_GRCh38_rna.fna, followed by adapter and low-quality trimming with Cutadapt v4.9 (Phred cutoff Q20; minimum read length 25 nt). Read quality before and after trimming was assessed with FastQC and summarized with MultiQC. Filtered reads were aligned to the Mus musculus reference genome GRCm38.p6 (RefSeq GCF_000001635.26; genomic FASTA GCF_000001635.26. Functional enrichment used clusterProfiler v4.16.0 with org.Mm.eg.db (v3.21.0). For Gene Ontology (GO; BP/CC/MF) over-representation analysis (ORA), the gene set comprised DE genes with |log2FC| ≥ 0.26 (≈20% change) and FDR < 0.05. Gene Set Enrichment Analysis (GSEA) was performed in clusterProfiler on a pre-ranked vector (e.g., signed statistic based on log2FC), with 10,000 permutations, minGSSize = 10, maxGSSize = 500, and significance defined as FDR q < 0.25 (GSEA convention). Pathway analysis used KEGG (Kyoto Encyclopedia of Genes and Genomes) gene sets within the same framework and genome build. GO/GSEA/KEGG outputs are reported as normalized enrichment score (NES), FDR q-value, and leading-edge genes, with full ranked lists and enrichment tables provided in Appendix A.

### 2.9. RT-PCR and Quantitative PCR

Total RNA was prepared as described in Section 2.10. cDNA was synthesized using ReverTra Ace^®^ qPCR RT Master Mix (TOYOBO Co., Ltd., Osaka, Japan) with random hexamers/oligo(dT) primers.

Conventional RT-PCR. Targets (*Trpm2*, *Nppa* (ANP), *Npr1* (NPR-A)) and housekeeping genes (*Gapdh*) were amplified using TOYOBO PCR reagents (e.g., KOD One™ PCR Master Mix, TOYOBO). Products were separated on agarose gels and visualized using SYBR™ Safe (Thermo Fisher Scientific) to confirm single bands at the expected sizes (representative gels shown in Figure 1).

qPCR. Quantitative PCR was performed with THUNDERBIRD^®^ SYBR^®^ qPCR Mix (TOYOBO) on a LightCycler^®^ 480 system (Roche Diagnostics, Basel, Switzerland). Targets were *Nppa*, and *Npr1*, and *Myl7* (used as an atrial-specific marker for normalization where indicated). *Gapdh* served as the reference gene. Reactions were run in technical triplicates; specificity was verified by melting-curve analysis. Relative expression was calculated using the 2^ΔΔCt^ method. Primer sequences and expected amplicon sizes are listed in Appendix A.

### 2.10. ELISA for ANP

At the endpoint of ISO ± ANP treatment, plasma (collected into EDTA-coated tubes and centrifuged at 2000× *g*, 10 min, 4 °C) was prepared. ANP concentrations were measured using a Mouse ANP ELISA kit (Arbor Assays, Ann Arbor, MI, USA) according to the manufacturer’s instructions. Absorbance was read on a microplate reader (Infinite M200 microplate reader, Tecan Group Ltd., Männedorf, Switzerland), and concentrations were calculated from a standard curve (four-parameter logistic fit). Appropriate blanks and quality controls were included on each plate.

### 2.11. Data Presentation and Statistical Analysis

Data are shown as mean ± SEM; n denotes animals for in vivo studies and independent isolations for cell experiments. Two-group comparisons used unpaired two-tailed Student’s *t*-tests, and multi-group comparisons used one- or two-way ANOVA with Tukey’s post hoc tests as appropriate; correlations (e.g., plasma ANP vs. HW/BW) used Pearson’s r. A significance level of *p* < 0.05 was applied. Analyses were performed in GraphPad Prism 9 (GraphPad Software, San Diego, CA, USA) and OriginPro (OriginLab, Northampton, MA, USA). Assumptions for parametric tests were evaluated prior to analysis. Normality was assessed using the Shapiro–Wilk test and equality of variances was examined using Levene’s test. All datasets met the assumptions required for parametric testing; therefore, the statistical methods described above (*t*-tests, ANOVA with Tukey’s post hoc tests, and Pearson’s correlation) were applied without the need for non-parametric alternatives. The exact statistical test used for each analysis, together with n values and *p*-values, is reported in the figure legends.

## 3. Results

### 3.1. TRPM2 Is Enriched and Functionally Active in Atrial Myocytes

RT–PCR detected Trpm2 transcripts in atria from wild-type (WT) hearts, whereas the band was absent in TRPM2^−/−^ samples. In ventricular tissue, Trpm2 was not detectable under our RT–PCR conditions or, if present, near the threshold of detection (Figure 1A, Appendix A). Consistent with these gel-based observations, quantitative PCR demonstrated that *Trpm2* mRNA expression was markedly higher in atria than in ventricles. *Gapdh*-normalized qPCR showed significantly elevated *Trpm2* levels in WT atria compared with KO atria (Welch’s *t*-test, *p* < 0.05), and ventricular tissue from WT mice displayed only very low but detectable *Trpm2* expression, whereas KO samples were at background levels (Appendix A). When normalized to the atrial marker *Myl7*, the relative abundance of *Trpm2* in atria appeared even more pronounced, further supporting the notion that *Trpm2* is preferentially enriched in atrial myocardium (Figure 1A, right). Consistently, in ventricular myocytes, ADPr failed to evoke a discernible current in either genotype (Appendix A).

In isolated atrial myocytes, intracellular ADPr (ADPr, 0.5 mM) elicited a time-dependent whole-cell current in WT cells, whereas currents remained minimal in WT cells dialyzed without ADPr and in TRPM2^−/−^ cells irrespective of ADPr (Figure 1B). The ADPr-evoked current showed an approximately linear I–V relationship (Figure 1C). The increase in current density (ΔI) was significantly larger in WT + ADPr than in the other three conditions, which were near baseline (Figure 1D, *p* < 0.05). Consistently, exposure to H_2_O_2_ increased fura-2 ratios in WT atrial myocytes, whereas responses were markedly blunted in TRPM2^−/−^ cells (Figure 1E,F, *p* < 0.05).

We did not detect ADPr/H_2_O_2_-activated currents in ventricular cardiomyocytes, consistent with our finding of barely detectable levels of TRPM2 channel expression in these cells.

### 3.2. TRPM2 Deficiency Aggravates ISO-Induced Systolic Dysfunction, Hypertrophy, and Fibrosis

Echocardiography revealed typical M-mode patterns across genotypes at baseline; during ISO administration, the contraction amplitude progressively decreased, most prominently in TRPM2^−/−^ + ISO (Figure 2A). Fractional shortening (FS%) declined with ISO in both genotypes, but the reduction was significantly greater in TRPM2^−/−^ mice (Figure 2B). The magnitude of ISO-induced reduction in FS% in TRPM2^−/−^ mice corresponded to a mean difference of −5.3%, with a 95% confidence interval ranging from −9.5% to −1.1%. Consistent indices are shown in Appendix A: ejection fraction (EF) (Appendix A), heart rate (HR) (Appendix A), and left-ventricular end-systolic diameter (LVESD) (Appendix A), the latter increasing during ISO exposure. In contrast, left-ventricular end-diastolic diameter (LVEDD) showed no material change among groups (Figure 2C), indicating that the early impairment was largely systolic rather than dilatational.

Consistent with functional deterioration, heart weight/body weight (HW/BW) increased after ISO, with a larger gain in TRPM2^−/−^ than in WT (Figure 2D). The ISO-induced increase in HW/BW in TRPM2^−/−^ mice showed a mean difference of +0.89 g/g compared with WT, with a 95% confidence interval from −0.01 to +1.79. Histological examination supported these findings. Hematoxylin–eosin (HE)-stained sections showed apparent myocardial wall thickening after ISO in both genotypes, with TRPM2^−/−^ + ISO hearts exhibiting a more prominent degree of tissue enlargement (Figure 2E: HE). Masson’s trichrome (MT) staining further revealed extensive interstitial and perivascular collagen deposition in TRPM2^−/−^ + ISO mice, whereas fibrosis in WT + ISO remained comparatively mild (Figure 2E: MT, 2F).

Together, these results indicate that loss of TRPM2 exacerbates β-adrenergic stress–induced systolic dysfunction and pathological remodeling, with greater hypertrophy and fibrosis, while chamber size remains essentially unchanged over the observation period.

### 3.3. β-Adrenergic Stress Evokes an ANP Program in WT Atria That Is Blunted in TRPM2^−/−^

To probe upstream endocrine pathways, we performed bulk RNA-seq on atria from WT and TRPM2^−/−^ mice with or without ISO. Standard analyses (quality control, alignment, differential expression) were followed by pathway enrichment/GSEA. In WT atria, ISO upregulated gene sets related to natriuretic peptide signaling and secretory granule/exocytosis, whereas these enrichments were attenuated in TRPM2^−/−^ atria (Figure 3A,B, overview of typical analyses). As a focused visualization, Figure 3A is organized in two complementary heatmaps to identify TRPM2-dependent nodes within the ANP/secretory program. Left (ISO effect in WT): log_2_ fold-change for WT + ISO vs. WT highlights genes induced by ISO in WT (red = up, blue = down). Right (ΔISO in KO–WT): the difference in ISO effects defined as ΔISO = (log2 (KO + ISO/KO)) − (log2(WT + ISO/WT)) so that negative values (purple) denote attenuation of the ISO response in KO relative to WT. By reading the two panels together, candidates for TRPM2-mediated regulation are genes that are red on the left (ISO-induced in WT) and blue on the right (ISO induction reduced in KO). Within the ANP-focused gene set (*Nppa*/*Nppb*, *Npr*1–3, *Corin*, *Furin*/*Pcsk* family, and vesicle-exocytosis components such as *Vamp*/*Snap*/*Syt*), this pattern includes *Nppa* (and to a lesser extent *Nppb*) together with selected processing/vesicle-fusion genes, consistent with a TRPM2-linked ANP granule/exocytosis module. Heatmaps display row-wise z-scores of log_2_[TPM + 1] for visualization; statistical testing and orthogonal validation are provided by qPCR/ELISA in Figure 3B–D. For the matched 1-week cohort used for transcriptomics, representative histology and quantitative cardiac morphology are provided in Appendix A (HE/MT images, HW/BW, and fibrosis%).

Consistent with the transcriptomic signal, qPCR confirmed that *Nppa* (ANP) expression rose markedly in WT + ISO, but the induction was significantly blunted in TRPM2^−/−^ + ISO (Figure 3B). In plasma, ANP concentrations increased with ISO in WT and were lower in TRPM2^−/−^ under the same stress (Figure 3C). By contrast, atrial Npr1 (NPR-A) expression showed no material reduction in TRPM2^−/−^ (Figure 3D), indicating that the impaired endocrine response is not due to downregulation of the ANP receptor.

Across individual animals, plasma ANP showed an inverse relationship with HW/BW in WT during ISO exposure, whereas no clear inverse trend was observed in TRPM2^−/−^ (Figure 3E). Together, these data indicate that, under β-adrenergic stress, TRPM2 contributes to the induction of the atrial ANP program and to the elevation of circulating ANP, and that loss of TRPM2 limits this cardioprotective endocrine response.

### 3.4. Exogenous ANP Mitigates ISO-Induced Dysfunction and Remodeling in TRPM2^−/−^ Mice

To test whether the impaired endocrine (ANP) response contributes to the aggravated phenotype, we investigated how treatment with ANP (400 µg/kg/day, s.c., twice daily for 7 days) affects ISO-exposed TRPM2^−/−^ mice. Mice treated with ANP showed improved systolic function compared with those that only received the vehicle control. FS% and EF% measures were also significantly higher in those that received ANP (Figure 4A–C). By contrast, chamber dimensions (LVEDD/LVESD) and heart rates were not affected by ANP treatment (Figure 4D–F). As for histological and morphometric measures, our data show that ANP modestly lowered HW/BW (Figure 4G) and attenuated histological remodeling in ISO-treated TRPM2^−/−^ mice. Histological examination supported this effect: HE sections demonstrated pronounced myocardial wall thickening and tissue enlargement in ISO-treated TRPM2^−/−^ mice, whereas ANP administration visibly mitigated this ISO-evoked enlargement (Figure 4H, HE). MT staining likewise revealed dense interstitial and perivascular collagen deposition in TRPM2^−/−^ + ISO hearts, while ANP markedly reduced collagen accumulation and preserved myocardial architecture (Figure 4H, MT). Morphometric quantification confirmed that % fibrosis was indeed reduced in the presence of ANP (Figure 4I; *p* < 0.05). Overall, these results indicate that augmenting ANP signaling partially rescues ISO-induced systolic impairment and pathological remodeling in mice lacking TRPM2.

### 3.5. Ventricular Cardiomyocyte Hypertrophy Is Induced by ISO and Attenuated by ANP Irrespective of TRPM2

The finding that WT mice were able to mount a strong ANP response to ISO, whereas TRPM2^−/−^ mice were unable to do so, suggests that TRPM2 plays a role in the induction of ANP. Given that atria are the major source of cardiac ANP, and the fact that we detected substantial TRPM2 expression only in atria, suggests that the effect of TRPM2 on ANP production may be limited to the atria. To test this hypothesis, we investigated the effects of ISO and ANP treatment on cultured neonatal ventricular myocytes from both WT and TRPM2^−/−^ mice.

Indeed, our results show comparable responsiveness of WT and TRPM2^−/−^ to ISO and ANP exposure. As shown morphologically in Figure 5A, ISO stimulation produced a time-dependent hypertrophic response in both genotypes. Phalloidin staining revealed a progressive enlargement of the projected cell area, with cortical F-actin clearly outlining the expanded cell perimeter during ISO-evoked hypertrophic spreading at 24–72 h. These qualitative changes corresponded closely to the quantitative increase in CSA observed in both WT and TRPM2^−/−^ cells (Figure 5), and baseline CSA did not differ by genotype.

In the ANP intervention experiments (Figure 5C), ISO-treated myocytes showed the expected hypertrophic enlargement, visualized as increased cell spread area outlined by phalloidin-labeled cortical F-actin. Co-treatment with ANP (0.1 μM) significantly—but incompletely—reduced this ISO-induced increase in CSA in both WT and TRPM2^−/−^ myocytes (Figure 5C,D; *p* < 0.05 vs. ISO). Correspondingly, ANP-treated cells exhibited modestly smaller projected areas than ISO-only cells, although CSA values generally remained above control levels. Thus, ventricular ANP responsiveness appears preserved in the absence of TRPM2, consistent with the interpretation that the exacerbated in vivo remodeling in TRPM2^−/−^ mice primarily reflects an atrial (endocrine) deficit in ANP induction rather than altered ventricular sensitivity to ANP.

## 4. Discussion

This study reveals a previously unrecognized role of the TRPM2 channel in regulating cardiac endocrine function and protecting against stress-induced pathological remodeling. Known as a cellular sensor of oxidative stress and metabolic state, TRPM2 mediates the influx of Ca^2+^ and other cations in response to oxidative and metabolic stimuli (ADPr/H_2_O_2_; Figure 1B–F; see also TRPM2 activation by ADPr/oxidants in [14,15,16,17]). It is expressed in various cell types and tissues, including the heart, where it was shown to have both protective and detrimental effects [23,27]. The protective effect was attributed to TRPM2 serving as the influx pathway for Ca^2+^ required to maintain mitochondrial bioenergetics integrity after I/R injury [24,25].

Here, we show a novel cardioprotective mechanism for TRPM2, which is required for stress-evoked secretion of ANP, a key cardioprotective hormone. Loss of TRPM2 impairs ANP production under β-adrenergic stress, leading to exacerbated cardiac hypertrophy and fibrosis (Figure 2, Appendix A). Importantly, we found that the addition of exogenous ANP restores the cardioprotective response in TRPM2^−/−^ mice (Figure 4). This result strongly supports a crucial link between TRPM2 activity, stress-induced ANP secretion, and cardiac remodeling outcomes. Under our assay conditions, TRPM2 expression was predominantly detected in atria rather than in ventricles (Figure 1A, Appendix A). It is noteworthy that ANP is also produced primarily in the atria by atrial cardiomyocytes in response to hemodynamic and mechanical stress. This increased ANP production is believed to be a protective hormonal response to maintain healthy myocardium and cardiovascular homeostasis. The ANP release is a regulated Ca^2+^-dependent granule exocytotic process involving SNAREs on large dense-core vesicles [28,29]. Several ion channels and G-protein coupled receptors (GPCRs) are thought to regulate ANP secretion, although the influx pathway for the Ca^2+^ signal has not been identified.

Our studies comparing gene expression in WT and TRPM2^−/−^ atria with or without ISO using bulk RNA-seq revealed activation of secretory/exocytotic pathways in WT + ISO, with attenuated induction in TRPM2^−/−^ + ISO, and a blunted upregulation of the ANP program (e.g., *Nppa*) in the absence of TRPM2 (Figure 3A,B). The ANP receptor gene, Npr1 (NPR-A), did not appear to be affected by the absence of TRPM2. Its expression was maintained or just modestly increased under ISO in TRPM2^−/−^ atria. These results potentially reflect a compensatory/negative-feedback response to reduced ANP availability, and are consistent with our qPCR data (Figure 3D). Although ANP expression is normally upregulated during cardiac hypertrophy, TRPM2^−/−^ mice failed to show this expected induction under ISO stress (Figure 3E). This atypical transcriptional response indicates that TRPM2 is required not only for ANP secretion but also for proper stress-induced upregulation of the *Nppa* gene. These gene expression studies, combined with electrophysiology (Figure 1B–D) and Ca^2+^ imaging data (Figure 1E,F) on WT and TRPM2^−/−^ atrial cardiomyocytes, collectively identify TRPM2 as an upstream component linking β-adrenergic stress to endocrine (ANP) output. In heart atria, TRPM2 has a specialized endocrine role distinct from that of more broadly expressed cardiac ion channels, a conclusion further supported by our qPCR data (Figure 1A, Appendix A) showing clear atrial enrichment of *Trpm2* expression. Our data support a model in which TRPM2 supplies the stress-evoked Ca^2+^ input upstream of regulated ANP exocytosis. We attempted pilot measurements of ANP secretion from isolated atrial preparations stimulated with H_2_O_2_; however, the secreted ANP levels were consistently near the assay’s detection limit, preventing reliable quantification. Therefore, these preliminary data were not included. The plausibility of TRPM2 as a Ca^2+^-dependent regulator of ANP secretion is further supported by its established role in stimulus–secretion coupling in endocrine cells, including reports that TRPM2-mediated Ca^2+^ entry facilitates glucose-induced insulin release in pancreatic β-cells [30]. Together with classic findings that atrial ANP secretion is Ca^2+^ dependent and suppressed by Gd^3+^—a non-selective blocker of mechanosensitive cation channels that also inhibits TRPM2 currents [31,32,33,34,35]—our results fit within a broader framework in which TRPM2 couples metabolic/oxidative cues to regulated peptide secretion. This interpretation is compatible with recent evidence that the regulated exocytic machinery controls ANP release in cardiomyocytes [36,37] and with reports that NPR-A expression is downregulated by ligand/cGMP [38], providing a rationale for higher *Npr1* when ANP is limited in TRPM2 deficiency.

Beyond the atrial context, there is precedent for TRPM2 coupling metabolic/oxidative cues to secretory output in endocrine and immune cells. In pancreatic β-cells, TRPM2 is co-expressed with insulin, and its activity is potentiated by physiological warmth and cyclic ADPr; mild heating elevates cytosolic Ca^2+^ and promotes insulin release, an effect diminished by TRPM2 knockdown [30]. In macrophages, TRPM2 forms part of a redox–ADPr negative-feedback loop that restrains cytokine production and secretion upon inflammatory polarization, with TRPM2 deficiency augmenting the output of IL-1α/IL-6/TNF-α output [39]. Together, these reports support a generalizable role for TRPM2 as a stress-responsive gate that links ADPr/redox signals to regulated secretion, lending independent plausibility to our conclusion that TRPM2 operates upstream of stress-evoked ANP release in atrial cardiomyocytes.

In clinical practice, short-term infusion of recombinant human ANP (carperitide) is used for acute decompensated heart failure and produces prompt hemodynamic unloading—reductions in pulmonary capillary wedge pressure and systemic vascular resistance with increased stroke volume index—together with natriuresis/diuresis and an acceptable safety profile (82% clinical improvement in a 3777-patient registry; hypotension the main adverse event) [40,41,42]. While the short-term hemodynamic and decongestive effects are consistent, the longer-term clinical benefit remains uncertain. Extensive observational analyses and a recent meta-analysis show no apparent mortality reduction and, in some cohorts, higher in-hospital mortality [43,44,45]. In our study, exogenous ANP rescued ISO-induced dysfunction, hypertrophy, and fibrosis in TRPM2^−/−^ mice (Figure 4), supporting the idea that when endogenous ANP release is insufficient—such as during acute sympathetic/oxidative stress—ANP supplementation can restore a cardioprotective endocrine response. These findings raise the possibility that patients with impaired atrial TRPM2–ANP signaling might particularly benefit from early augmentation of the ANP axis (e.g., ANP administration, neprilysin inhibition, or future TRPM2-enhancing strategies), a hypothesis that warrants prospective testing.

Given the established view of the heart as an endocrine organ governed by the natriuretic peptide system [12,46,47], our findings extend TRPM2 beyond its classical roles in oxidative-stress sensing and I/R paradigms [19,24,25,48,49,50] by highlighting an atrial endocrine function. From a translational standpoint, approaches that enhance TRPM2-dependent signaling in atrial cells or otherwise amplify the ANP axis could help boost ANP output and mitigate pathological remodeling in heart failure or hypertension.

This work focuses on atrial cardiomyocytes, positioning TRPM2 as part of the ANP granule/regulated exocytosis mechanism. While functional activation (Figure 1) is consistent with plasma-membrane localization, we did not demonstrate co-localization with ANP granules or molecular coupling to the exocytic machinery. The in vivo gating dynamics of atrial TRPM2 under ISO—including possible ROS/ADPr microdomains—remain undefined, and it is unknown whether these atrial findings generalize to pressure overload or myocardial infarction. Because conventional TRPM2 knockout mice were used, we cannot exclude the possibility that TRPM2 deficiency affects the secretion of mediators other than ANP, as several exocytosis-related genes were downregulated in TRPM2^−/−^ atria. To address this possibility, we examined representative classical peptide hormone genes detectable in our RNA-seq dataset. Both Glucagon gene (Gcg) (logFC −0.349, *p* = 0.685) and Arginine vasopressin gene (Avp) (logFC 0.664, *p* = 0.744) were expressed but showed no significant changes, indicating that major endocrine transcripts remained largely unaffected under our experimental conditions. However, *Nppa* expression was selectively reduced, and exogenous ANP fully rescued the hypertrophy and fibrosis phenotypes, indicating that impaired ANP induction and secretion are likely the dominant contributors to the exaggerated ISO response. While additional TRPM2-dependent secretory pathways may exist, our findings highlight the TRPM2–ANP axis as the principal mechanism linking β-adrenergic stress to cardioprotection. These considerations further emphasize that the primary defect in TRPM2^−/−^ mice arises upstream at the level of atrial ANP production rather than downstream ANP signaling. Notably, in neonatal ventricular myocytes, exogenous ANP attenuated ISO-induced hypertrophy independently of TRPM2 (Figure 5A–D), suggesting that TRPM2 is not required for downstream ANP actions in ventricles, but is essential upstream for stress-evoked ANP production in atria.

## 5. Conclusions

Together, these findings support a TRPM2–Ca^2+^–ANP axis in which atrial TRPM2 supplies the stress-evoked Ca^2+^ input that facilitates endocrine ANP output—rather than modulating ANP responsiveness per se. To our knowledge, this is the first report of TRPM2 as a regulator of atrial endocrine signaling and a gatekeeper of cardioprotective ANP release under β-adrenergic stress, suggesting a tractable therapeutic target for stress-induced cardiac dysfunction.

## Figures and Tables

**Figure 1 cells-15-00024-f001:**
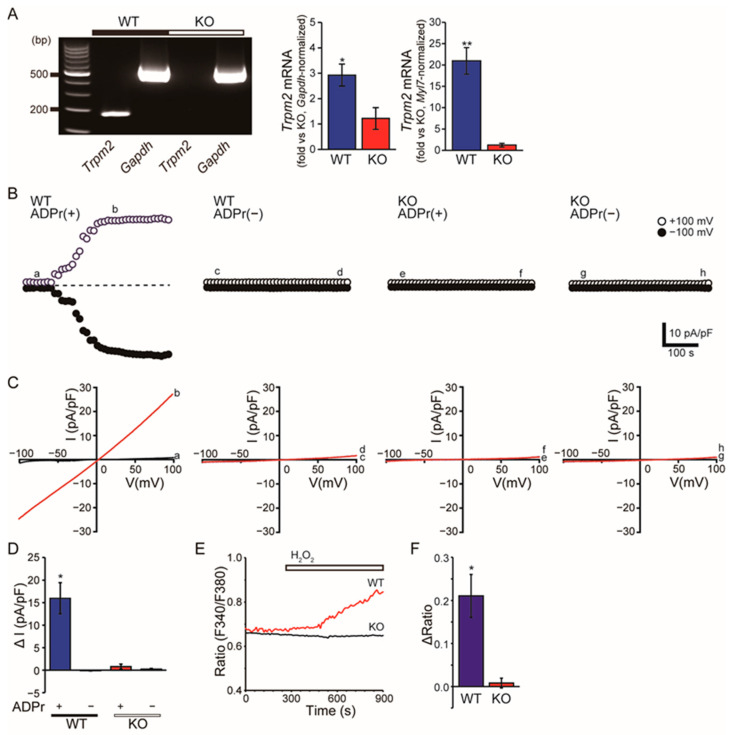
Atrial TRPM2 expression and TRPM2-dependent Ca^2+^ signals. (**A**) RT-PCR of *Trpm2* and *Gapdh* from atria in WT and TRPM2^−/−^ (KO) mice; representative gel images are shown. *Gapdh* is the loading control. Right: Quantification of *Trpm2* mRNA expression in WT and KO atria by real-time PCR normalized to *Gapdh* or to *Myl7*. Data are presented as mean ± SEM. Statistical comparisons were performed using Welch’s *t*-test (WT vs. KO: *p* < 0.05) (**B**) Whole-cell patch clamp from isolated atrial myocytes: time courses of currents at +100 mV (open symbols) and −100 mV (filled symbols) recorded with 0.5 mM ADP-ribose in the pipette (ADPr (+)) or without ADPr (ADPr (−)). The horizontal dotted line denotes the zero-current baseline. (**C**) Corresponding I–V relationships (ramps from +100 to −100 mV) at the indicated time points (a–h in panel (**B**)). (**D**) Summary of ADPr-activated current density (ΔI at +100 mV, pA/pF). Data are presented as mean ± SEM. Statistics evaluate within-genotype ADPr (+) vs. ADPr (−) using unpaired two-tailed Student’s *t*-tests. (n = 5–25) * *p* < 0.05 vs. ADPr (−). (**E**) Fura-2 ratiometric Ca^2+^ imaging in atrial myocytes during extracellular H_2_O_2_ (200 µM; bar) in Tyrode’s solution. (**F**) Quantified peak ΔRatio from (**E**). Data are mean ± SEM. Comparisons between genotypes were performed using unpaired two-tailed Student’s *t*-tests (n = 10–18) * *p* < 0.05, ** *p* < 0.01 vs. KO.

**Figure 2 cells-15-00024-f002:**
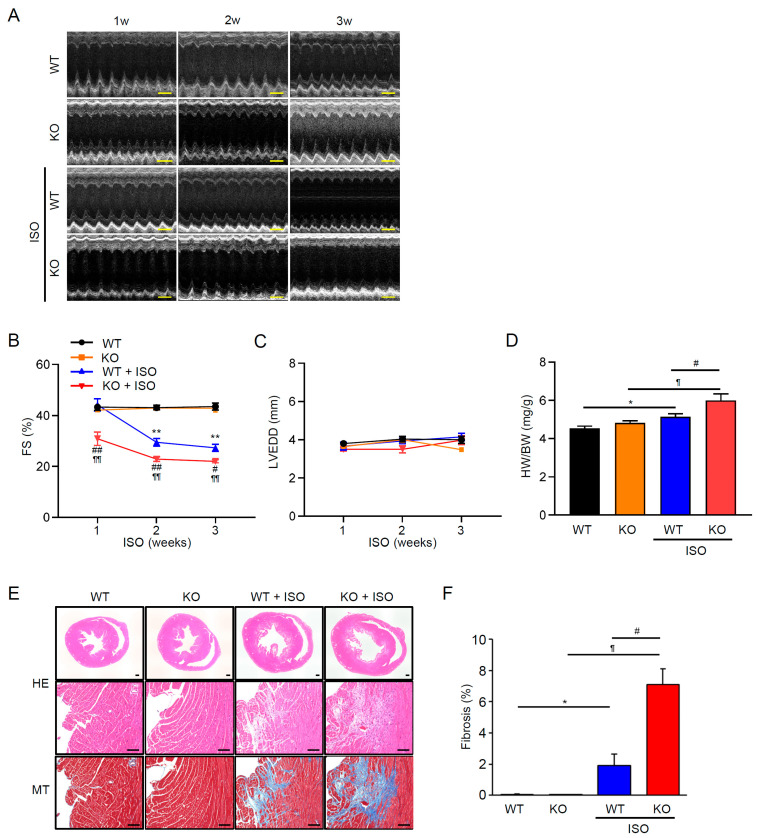
TRPM2 deficiency aggravates ISO-induced systolic dysfunction, hypertrophy, and fibrosis. (**A**) Representative M-mode echocardiograms (baseline, 1–3 weeks) in WT and TRPM2^−/−^ (KO) mice. Scale bar (horizontal), 200 ms. (**B**) Time course of fractional shortening (FS, %) Data are mean ± SEM (n = 5–6). (**C**) Left-ventricular end-diastolic diameter (LVEDD, mm) over time (n = 5–6). (**D**) Heart-weight–to–body-weight ratio (HW/BW) at endpoint (n = 6–8). (**E**) Histology of whole-heart sections and mid-ventricular myocardium. Hematoxylin–eosin (HE) for myocyte morphology; Masson’s trichrome (MT) for interstitial/perivascular collagen. Scale bars: 100 µm (upper panels), 500 µm (lower panels). (**F**) Quantification of fibrotic area (% of tissue). Data are mean ± SEM (n = 6). Dots represent individual animals. (**B**,**D**,**F**) Statistical analysis: FS% and LVEDD were compared across time using two-way ANOVA with Tukey’s post hoc test; HW/BW and fibrosis% were compared by one-way ANOVA with Tukey’s post hoc test. * *p* < 0.05, ** *p* < 0.01 WT vs. WT + ISO; ^¶^
*p* < 0.05, ^¶¶^ *p* < 0.01 KO vs. KO + ISO; ^#^
*p* < 0.05, ^##^
*p* < 0.01 WT + ISO vs. KO + ISO.

**Figure 3 cells-15-00024-f003:**
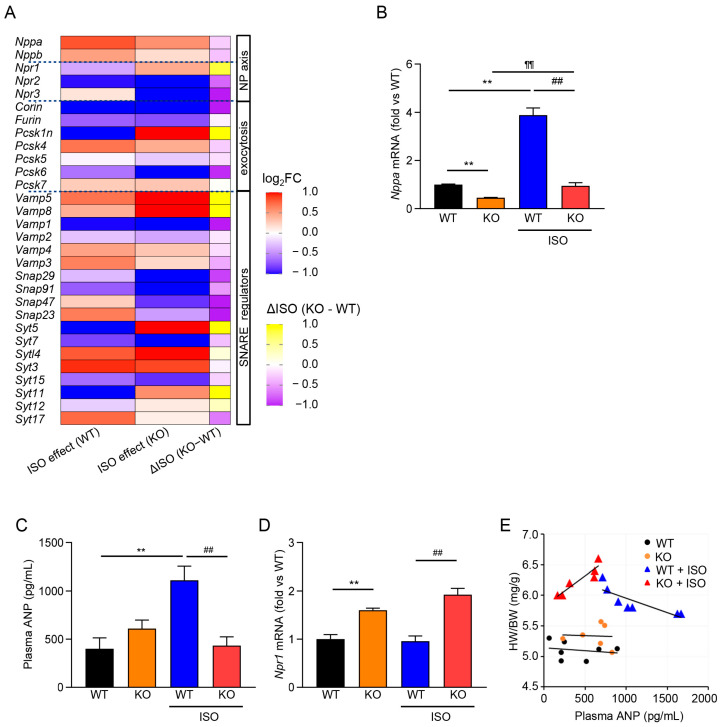
TRPM2 deficiency blunts the atrial ANP program under ISO. (**A**) Focused heatmap of ANP/secretory–exocytosis genes (*Nppa*, *Nppb*, *Npr1–3*, *Corin*, *Furin*/*Pcsk* family, *Vamp*/*Snap*/*Syt*) across WT, KO, WT + ISO, KO + ISO. Values are row-wise z-scores of log_2_(TPM + 1); genes are clustered by Euclidean distance. ISO effect (WT) = log_2_FC (WT + ISO vs. WT); ISO effect (KO) = log_2_FC (KO + ISO vs. KO); ΔISO (KO − WT) = ISO effect (KO) − ISO effect (WT). (**B**) qPCR of *Nppa* (ANP) mRNA in atria (relative to WT) (mean ± SEM, n = 5). (**C**) Plasma ANP by ELISA (mean ± SEM, n = 6–7). (**D**) Atrial *Npr1* (NPR-A) expression (mean ± SEM, n = 5). (**E**) Correlation between plasma ANP and HW/BW. Linear least-squares fits of HW/BW (mg·g^−1^) vs. plasma ANP (pg·mL^−1^): WT, y = −1.0 × 10^−4^ x + 5.14; WT + ISO, y = −5.0 × 10^−4^ x + 6.43; KO, y = −4.0 × 10^−5^ x + 5.36; KO + ISO, y = 1.0 × 10^−3^ x + 5.83. (**B**–**D**) Statistical testing: one-way ANOVA with Tukey’s post hoc test. ** *p* < 0.01 WT vs. WT + ISO; ^¶¶^ *p* < 0.01 KO vs. KO + ISO; ^##^ *p* < 0.01 WT + ISO vs. KO + ISO.

**Figure 4 cells-15-00024-f004:**
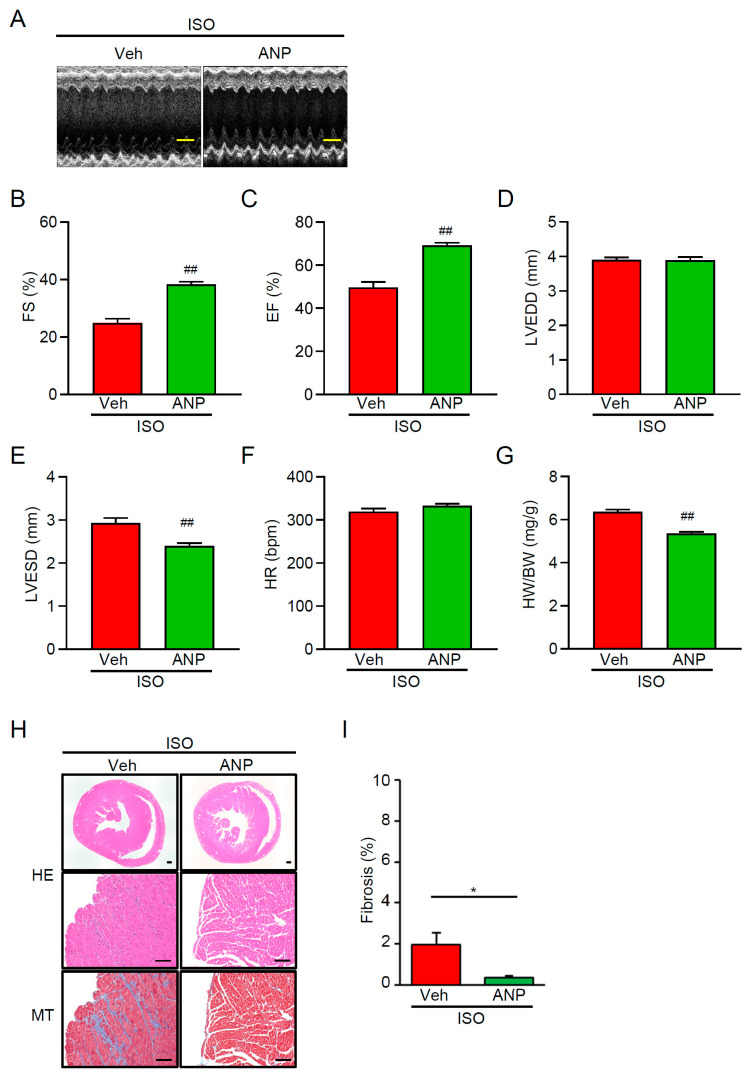
Exogenous ANP rescues dysfunction and remodeling in ISO-treated TRPM2^−/−^ hearts. TRPM2^−/−^ (KO) mice received ISO (30 mg·kg^−1^·day^−1^, i.p., 7 days) and were co-treated with vehicle or ANP (400 µg·kg^−1^, s.c., 7 days). (**A**) Representative M-mode echocardiograms (Vehicle vs. ANP). Scale bar (horizontal), 200 ms. (**B**,**C**) Fractional shortening (FS, %) and ejection fraction (EF, %) for Vehicle and ANP groups (mean ± SEM; n = 6). (**D**,**E**) Left-ventricular diameters (LVEDD, LVESD, mm) for Vehicle and ANP groups. (**F**,**G**) Heart rate (HR, bpm) and heart-weight to body-weight ratio (HW/BW) for Vehicle and ANP groups (mean ± SEM; n = 6). (**H**) Representative histology: H&E and Masson’s trichrome (MT) sections. Scale bars: 100 µm (upper panels), 500 µm (lower panels). (**I**) Quantification of fibrosis area (% LV) (mean ± SEM; n = 6). Statistical testing (**B**–**G**,**I**): unpaired two-tailed Student’s *t*-test. * *p* < 0.05 Veh + ISO vs. ANP + ISO; ^##^ *p* < 0.01 Veh + ISO vs. ANP + ISO.

**Figure 5 cells-15-00024-f005:**
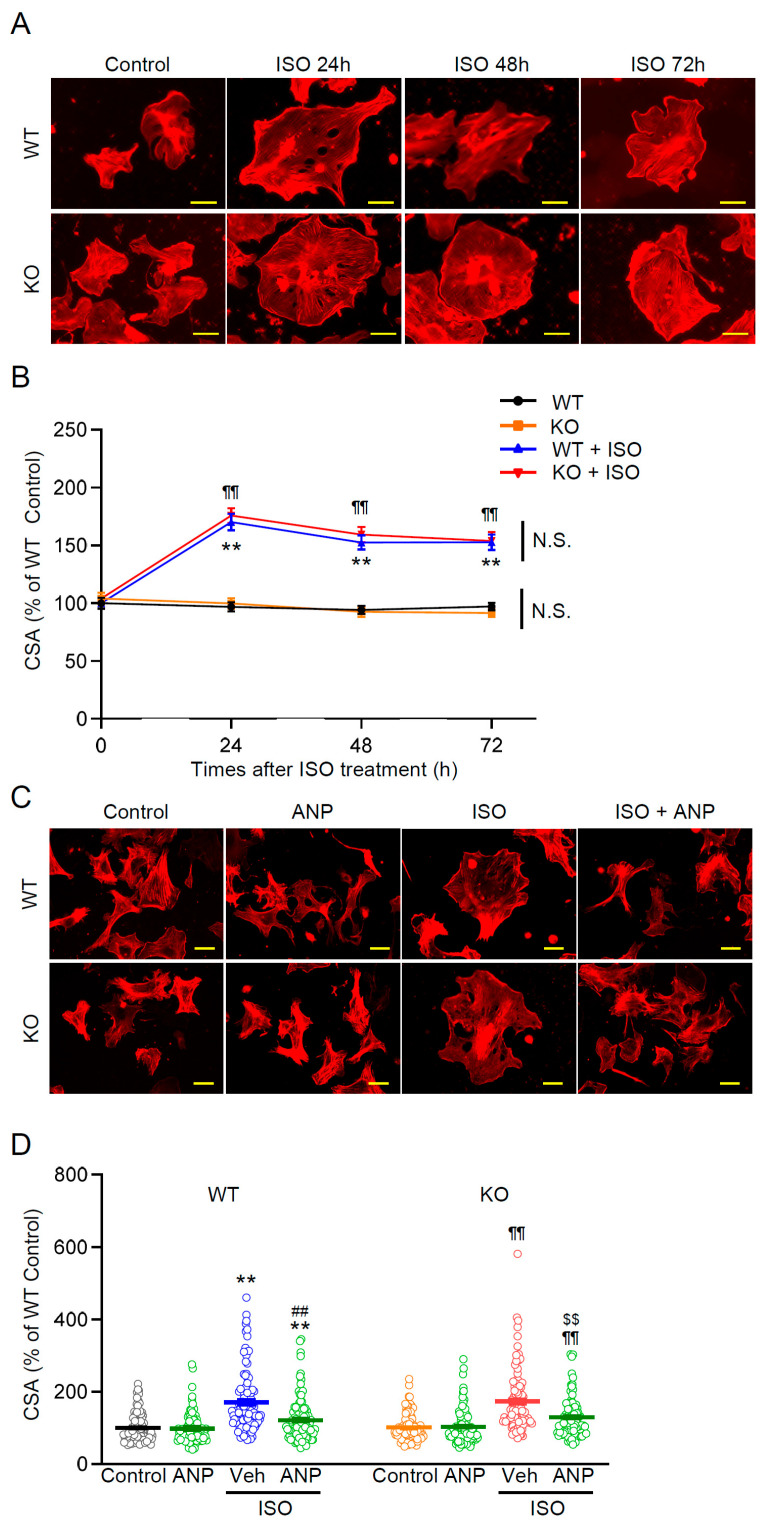
ISO-induced ventricular hypertrophy and its attenuation by ANP are comparable in WT and TRPM2^−/−^ cardiomyocytes. (**A**) Representative images of neonatal ventricular myocytes (phalloidin, F-actin) from WT and TRPM2^−/−^ (KO) mice under control conditions and after ISO (10 µM, 24–72 h). Scale bar, 50 µm. (**B**) Quantification of cell cross-sectional area (CSA) over 24–72 h, normalized to WT control; WT and TRPM2^−/−^ were compared at each time point (mean ± SEM, n =100–117). (**C**) Representative images of cells with or without ANP (0.1 µM, 24 h) in basal and ISO-treated conditions. Scale bar, 50 µm. (**D**) CSA quantification across the indicated conditions for both genotypes (mean ± SEM, n =101–108). (**B**,**D**): two-way ANOVA followed by Tukey’s post hoc test. ** *p* < 0.01 WT vs. WT + ISO ± ANP; ^¶¶^ *p* < 0.01 KO vs. KO + ISO ± ANP; ^##^ *p* < 0.01 WT + ISO + Veh vs. WT + ISO + ANP; ^$$^ *p* < 0.01 KO + ISO + Veh vs. KO + ISO + ANP; ^N.S.^ not significant.

## Data Availability

The RNA-seq datasets generated in this study have been submitted to the NCBI Sequence Read Archive (SRA) under submission ID SUB15770329 and will be made publicly available upon acceptance. Final accession numbers will be provided in the published article. Source data underlying all figures and analysis scripts are available from the corresponding author upon reasonable request.

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
