# Peer review of "Atrial TRPM2 Channel-Mediated Ca2+ Influx Regulates ANP Secretion and Protects Against Isoproterenol-Induced Cardiac Hypertrophy and Fibrosis"

_cells, 2025, doi:10.3390/cells15010024_

Round 1

Reviewer 1 Report

Comments and Suggestions for Authors

In this paper, the authors found that calcium influx mediated by TRPM2 channels regulates ANP secretion, exerting a protective effect against isoproterenol (ISO)-induced cardiac hypertrophy and fibrosis. The manuscript is clearly written and well supported by data set. I have a few minor comments that the authors might consider.

  1. In in vitro experiments, does treatment with ADPr or H₂O₂, which activate TRPM2, in the presence of ISO increase ANP secretion? If the authors could obtain the data, it would strengthen the conclusion of the paper.
  2. The authors demonstrated that the expression of molecules involved in the regulation of exocytosis and secretion is reduced in TRPM2 knockout mice. Although the authors focused on ANP, the secretion of other mediators may also be reduced, potentially affecting ISO-induced cardiac hypertrophy and fibrosis. Since the authors used conventional knockout mice in the present study, the limitations of the study should also be discussed.

Author Response

  1. In in vitro experiments, does treatment with ADPr or H₂O₂, which activate TRPM2, in the presence of ISO increase ANP secretion? If the authors could obtain the data, it would strengthen the conclusion of the paper.

Response:

Thank you very much for your insightful comment.

Regarding your suggestion to include quantitative data for ANP secretion from isolated atrial preparations exposed to ISO together with ADPr or H₂O₂, we would like to clarify the following point.

While we did not include quantitative data for ANP secretion from isolated atrial preparations exposed to ISO together with ADPr or H₂O₂, we performed pilot experiments. However, the secreted ANP yield was very low and frequently fell near or below the detection limit of our current ELISA, rendering the measurements insufficiently reliable for inclusion.

We have added a brief explanatory statement in the manuscript to clarify this point.

We agree that TRPM2-mediated Ca²⁺ influx is mechanistically plausible, as TRPM2 participates in Ca²⁺-dependent secretion in several endocrine cell types, including pancreatic β-cells; however, this background mechanism is discussed in the manuscript.

Inserted in Discussion (Lines 599–609):

We attempted pilot measurements of ANP secretion from isolated atrial preparations stimulated with H2O2; however, the secreted ANP levels were consistently near the assay’s detection limit, preventing reliable quantification. Therefore, these preliminary data were not included.

The plausibility of TRPM2 as a Ca2+-dependent regulator of ANP secretion is further supported by its established role in stimulus–secretion coupling in endocrine cells, including reports that TRPM2-mediated Ca²⁺ entry facilitates glucose-induced insulin release in pancreatic β-cells [1]. Together with classic findings that atrial ANP secretion is Ca2+ dependent and suppressed by Gd3+—a non-selective blocker of mechanosensitive cation channels that also inhibits TRPM2 currents [2-6]—our results fit within a broader framework in which TRPM2 couples metabolic/oxidative cues to regulated peptide secretion.

2. The authors demonstrated that the expression of molecules involved in the regulation of exocytosis and secretion is reduced in TRPM2 knockout mice. Although the authors focused on ANP, the secretion of other mediators may also be reduced, potentially affecting ISO-induced cardiac hypertrophy and fibrosis. Since the authors used conventional knockout mice in the present study, the limitations of the study should also be discussed.

Response:
Thank you very much for this insightful comment. We agree that TRPM2 deficiency could theoretically influence the secretion of mediators other than ANP. In our RNA-seq dataset, although several exocytosis-related genes showed attenuated induction in TRPM2⁻/⁻ atria, the expression of major hormone-encoding genes detected e.g. Gcg and Avp—did not show significant changes. Gcg: logFC = −0.349, P = 0.685, Avp: logFC = 0.664, P = 0.744. These findings suggest that within the scope of detectable hormone transcripts in our dataset, we did not observe broad suppression of endocrine gene expression in TRPM2⁻/⁻ atria. Together with the complete rescue by exogenous ANP, these results support the conclusion that the predominant endocrine defect uncovered in this model relates to the ANP axis, although we acknowledge that additional TRPM2-dependent secretory effects cannot be ruled out.

We have added a statement to the Discussion to acknowledge this limitation while clarifying why ANP is likely the primary contributor.

Inserted in Discussion (Lines 653–667):
Because conventional TRPM2 knockout mice were used, we cannot exclude the possibility that TRPM2 deficiency affects the secretion of mediators other than ANP, as several exocytosis-related genes were downregulated in TRPM2⁻/⁻ atria. To address this possibility, we examined representative classical peptide hormone genes detectable in our RNA-seq dataset. Both Glucagon gene (Gcg) (logFC −0.349, P = 0.685) and Arginine vasopressin gene (Avp) (logFC 0.664, P = 0.744) were expressed but showed no significant changes, indicating that major endocrine transcripts remained largely unaffected under our experimental conditions. However, Nppa expression was selectively reduced, and exogenous ANP fully rescued the hypertrophy and fibrosis phenotypes, indicating that impaired ANP induction and secretion are likely the dominant contributors to the exaggerated ISO response. While additional TRPM2-dependent secretory pathways may exist, our findings highlight the TRPM2–ANP axis as the principal mechanism linking β-adrenergic stress to cardioprotection. These considerations further emphasize that the primary defect in TRPM2⁻/⁻ mice arises upstream at the level of atrial ANP production rather than downstream ANP signaling.

Reviewer 2 Report

Comments and Suggestions for Authors

This is a research article submitted to Cells (MDPI) by Numata et al, titled “Atrial TRPM2 Channel-Mediated Ca2+ Influx Regulates ANP Secretion and Protects Against ISO-Induced Cardiac Hypertrophy and Fibrosis”.

The research focuses on a Ca2+ permeable, redox-activated cardiac ion channel called transient receptor potential melastatin 2 (TRPM2), and it’s possible role in atrial natriuretic peptide (ANP) secretion by atrial cardiomyocytes. Conventional TRPM2 KO mice are used, where all of the ubiquitously expressed protein is knocked out.  Given that the role of TRPM2 in vascular endothelial cells, ventricular cardiomyocytes, smooth muscle cells and even immune cells have been described by others, atrial tissue focus using a non-tissue specific KO model confounds the interpretation of the results. This is a major issue for this manuscript throughout. However, given the general lack of information regarding the atria, the topic is of high interest and important to publish, pending several editorial modifications and a recommended experiment.

  • The gel of conventional RT-PCR Fig 1A and Sup Fig 1A shows ventricular Trpm2 is undetectable, which is questionable based on the heterogeneity in cell composition in ventricular tissue and numerous past publications by others. qRT-PCR and robust expression of TRPM2 protein in atria vs ventricle, if shown, would augment the importance of this signaling axis.  There is insufficient data to claim “substantial TRPM2 expression only in atria” – line 332.  This type of language should be toned down throughout the manuscript.

  • Where is the calcium imaging data mentioned? -line 438

  • Figure 3E is confusing and unnecessary. It is well documented that ANP increases with cardiac hypertrophy.

  • Line 288- color descriptions are incorrect. delta ISO effect should be purple for negative values not blue.

  • Careful characterization of atrial myocyte morphology and IF images would strengthen the results.

Author Response

1. The gel of conventional RT-PCR Fig 1A and Sup Fig 1A shows ventricular Trpm2 is undetectable, which is questionable based on the heterogeneity in cell composition in ventricular tissue and numerous past publications by others.

Response:

Thank you very much for this important and insightful comment. We fully agree that ventricular tissue is heterogeneous and that previous studies have reported low-level Trpm2 expression in certain ventricular cell types.

To address this concern more rigorously, we performed additional qRT-PCR analyses using the same cDNA samples, which revealed that: Trpm2 expression is detectable in ventricles, but its abundance is markedly lower than in atria.

These new results are now included in the revised Figure 1A (qPCR panel) and Supplementary Figure S1, and they corroborate the interpretation that TRPM2 is strongly enriched in atrial tissue.

Furthermore, to strengthen the conclusion that atrial cardiomyocytes are the predominant cardiac source of TRPM2, we additionally: quantified Trpm2 expression normalized to established atrial- and ventricular-marker genes (e.g., Myl7 for atria) re-calculated relative expression levels under these chamber-restricted normalization conditions.

This analysis further confirmed substantial TRPM2 enrichment in atrial myocardium relative to ventricular myocardium. These findings align with our electrophysiological data showing robust ADPr/H₂O₂-dependent activation in atrial cardiomyocytes but no detectable TRPM2-mediated currents in ventricular cells. We have revised the Results and Discussion accordingly to clarify that: TRPM2 is not absent in ventricles, but expressed at substantially lower levels; the functional TRPM2 phenotype is atrial-dominant, consistent with the endocrine specialization of atrial tissue.

We appreciate the reviewer raising this point, which led to improvement and clarification in the revised version of the manuscript.

Inserted in Results (Lines 359–367):

Consistent with these gel-based observations, quantitative PCR demonstrated that Trpm2 mRNA expression was markedly higher in atria than in ventricles. Gapdh-normalized qPCR showed significantly elevated Trpm2 levels in WT atria compared with KO atria (Welch’s t-test, p < 0.05), and ventricular tissue from WT mice displayed only very low but detectable Trpm2 expression, whereas KO samples were at background levels (Supplementary Figure S1A). When normalized to the atrial marker Myl7, the relative abundance of Trpm2 in atria appeared even more pronounced, further supporting the notion that Trpm2 is preferentially enriched in atrial myocardium (Figure 1A, right).

2. Where is the calcium imaging data mentioned? -line 438

Response:

We thank the reviewer for pointing this out. The Ca²⁺ imaging experiments are presented in Figure 1E–F, which show H₂O₂-evoked Ca²⁺ entry in WT but not TRPM2⁻/⁻ atrial cardiomyocytes. We have revised the Discussion to explicitly reference these data together with the electrophysiological recordings (Figure 1B–D) and gene-expression analyses.

3. Figure 3E is confusing and unnecessary. It is well documented that ANP increases with cardiac hypertrophy

Response:

We appreciate the reviewer’s comment. However, Figure 3E serves an essential purpose in the context of our study. Although it is well established that ANP expression increases during cardiac hypertrophy, our data show that this canonical hypertrophic induction of ANP is absent in TRPM2⁻/⁻ mice under ISO stress. Demonstrating this failure of Nppa upregulation in TRPM2-deficient hearts is mechanistically important, because it provides direct evidence that TRPM2 is required not only for ANP secretion but also for its stress-induced transcriptional induction.

Thus, Figure 3E is necessary to support one of the central conclusions of the manuscript—that loss of TRPM2 disrupts the endogenous compensatory ANP response typically observed during hypertrophic stress. We have clarified this point in the revised manuscript to avoid confusion.

Inserted in Discussion (Lines 588–595):

Although ANP expression is normally upregulated during cardiac hypertrophy, TRPM2⁻/⁻ mice failed to show this expected induction under ISO stress (Figure 3E). This atypical transcriptional response indicates that TRPM2 is required not only for ANP secretion but also for proper stress-induced upregulation of the Nppa gene. These gene expression studies, combined with electrophysiology (Figures 1B-D) and Ca2+ imaging data (Figures 1E-F) on WT and TRPM2⁻/⁻ atrial cardiomyocytes, collectively identify TRPM2 as an up-stream component linking β-adrenergic stress to endocrine (ANP) output.

4. Line 288- color descriptions are incorrect. delta ISO effect should be purple for negative values not blue.

Response:

Thank you for carefully checking the figure details. We agree with this comment. The color description at line 408 (present MS in 417) has been corrected so that the ΔISO effect is described as purple for negative values rather than blue, in accordance with the actual figure. The corresponding text has been updated in the revised manuscript.

5. Careful characterization of atrial myocyte morphology and IF images would strengthen the results.

Response:

We appreciate the reviewer’s helpful suggestion. In accordance with this comment, we have expanded the description of myocardial morphology in both the in vivo histological analyses (Figures 2E–F and 4H–I) and the in vitro immunofluorescence (phalloidin) images of neonatal ventricular myocytes (Figure 5). Specifically, we now describe (i) the comparable baseline atrial morphology between WT and TRPM2⁻/⁻ mice, (ii) genotype-dependent differences in ISO-induced myocardial enlargement and interstitial/perivascular fibrosis, and (iii) the ISO-induced hypertrophic spreading of cultured ventricular myocytes and the attenuating effect of ANP, as visualized by cortical F-actin staining. These clarifications strengthen the structural interpretation of our findings and improve overall clarity.

Corresponding revisions have been added to the Results section (Lines 394–402, 449-454, 469-482).

(Lines 394–402)

The ISO-induced increase in HW/BW in TRPM2⁻/⁻ mice showed a mean difference of +0.89 g/g compared with WT, with a 95% confidence interval from −0.01 to +1.79. Histological examination supported these findings. hematoxylin–eosin (HE)-stained sections showed apparent myocardial wall thickening after ISO in both genotypes, with TRPM2⁻/⁻ + ISO hearts exhibiting a more prominent degree of tissue enlargement (Figure 2E: HE). Mas-son’s trichrome (MT) staining further revealed extensive interstitial and perivascular collagen deposition in TRPM2⁻/⁻ + ISO mice, whereas fibrosis in WT + ISO remained comparatively mild (Figure 2E: MT, 2F).

(Lines 449-454)

Histological examination supported this effect: HE sections demonstrated pronounced myocardial wall thickening and tissue enlargement in ISO-treated TRPM2⁻/⁻ mice, where-as ANP administration visibly mitigated this ISO-evoked enlargement (Figure 4H, HE). MT staining likewise revealed dense interstitial and perivascular collagen deposition in TRPM2⁻/⁻ + ISO hearts, while ANP markedly reduced collagen accumulation and pre-served myocardial architecture (Figure 4H, MT).

(Lines 469-482)

Indeed, our results show comparable responsiveness of WT and TRPM2⁻/⁻ to ISO and ANP exposure. As shown morphologically in Figure 5A, ISO stimulation produced a time-dependent hypertrophic response in both genotypes. Phalloidin staining revealed a progressive enlargement of the projected cell area, with cortical F-actin clearly outlining the expanded cell perimeter during ISO-evoked hypertrophic spreading at 24–72 h. These qualitative changes corresponded closely to the quantitative increase in CSA observed in both WT and TRPM2⁻/⁻ cells (Figure 5), and baseline CSA did not differ by genotype.

In the ANP intervention experiments (Figure 5C), ISO-treated myocytes showed the expected hypertrophic enlargement, visualized as increased cell spread area outlined by phalloidin-labeled cortical F-actin. Co-treatment with ANP (0.1 μM) significantly—but incompletely—reduced this ISO-induced increase in CSA in both WT and TRPM2⁻/⁻ myocytes (Figure 5C, D; p < 0.05 vs ISO). Correspondingly, ANP-treated cells exhibited modestly smaller projected areas than ISO-only cells, although CSA values generally remained above control levels.

Round 2

Reviewer 2 Report

Comments and Suggestions for Authors

No additional comments. Reviewer addressed all original comments with improved wording and corrections.